# A Review on Tetrabromobisphenol A: Human Biomonitoring, Toxicity, Detection and Treatment in the Environment

**DOI:** 10.3390/molecules28062505

**Published:** 2023-03-09

**Authors:** Baoji Miao, Salome Yakubu, Qingsong Zhu, Eliasu Issaka, Yonghui Zhang, Mabruk Adams

**Affiliations:** 1Henan International Joint Laboratory of Nano-Photoelectric Magnetic Materials, School of Materials Science and Engineering, Henan University of Technology, Zhengzhou 450001, China; 2School of Environment and Safety Engineering, Jiangsu University, Zhenjiang 212013, China; 3School of Civil Engineering, National University of Ireland, H91 TK33 Galway, Ireland

**Keywords:** tetrabromobisphenol A, human biomonitoring, toxicity, treatment, brominated flame retardant

## Abstract

Tetrabromobisphenol A (TBBPA) is a known endocrine disruptor employed in a range of consumer products and has been predominantly found in different environments through industrial processes and in human samples. In this review, we aimed to summarize published scientific evidence on human biomonitoring, toxic effects and mode of action of TBBPA in humans. Interestingly, an overview of various pretreatment methods, emerging detection methods, and treatment methods was elucidated. Studies on exposure routes in humans, a combination of detection methods, adsorbent-based treatments and degradation of TBBPA are in the preliminary phase and have several limitations. Therefore, in-depth studies on these subjects should be considered to enhance the accurate body load of non-invasive matrix, external exposure levels, optimal design of combined detection techniques, and degrading technology of TBBPA. Overall, this review will improve the scientific comprehension of TBBPA in humans as well as the environment, and the breakthrough for treating waste products containing TBBPA.

## 1. Introduction

Brominated flame retardants (BFR) have been extensively applied to reduce the flammability of some commercial products such as furniture, circuit boards, textiles, polystyrene foams, epoxy resins, and padding materials [1,2,3], because of their potency and ability to meet safety standards. These BFRs include chemicals, such as chlorine, bromine, and phosphorus. Currently, tetrabromobisphenol A (TBBPA) is among the most utilized flame retardants in the industry globally [4,5], which is classified as an endocrine disruptor. Endocrine disruptors are chemical compounds which inhibit the activities of natural hormones in the body, such as secretion, binding, transport, synthesis, action, or elimination responsible for maintaining reproduction, homeostasis, behavior, or development. Numerous concerns have been raised about human exposure to these disruptors, primarily because of the assumed detrimental effect they pose to human health [1,6,7].

Owing to the high volume of production and potential human exposures, the toxicity of TBBPA has been investigated in a number of experimental studies. Research carried out on TBBPA has revealed less than 4% of its particles in dust are respirable and less than 10 µm can be absorbed from the lungs for systemic circulation after inhalation [8]. However, based on the physicochemical properties of TBBPA, its absorption via dermal exposure is expected to be poor. Data from an in vitro study, conducted with human skin, showed that less than 1% of the administered dose was absorbed dermally [9]. Human samples and rat strains have been studied for the metabolism and toxicokinetic action of TBBPA, which has confirmed that TBBPA can be absorbed from the gastrointestinal tract and rapidly eliminated after conjugation or Phase II metabolism to more water-soluble metabolites [10]. TBBPA has low (F < 0.05) systemic bioavailability due to its extensive hepatic bio-transformation to glucuronides and sulfates, which are excreted from the liver predominantly with bile as a result of their high molecular weight. Knudsen’s group [11] observed in their study that excretion was delayed only after a single dose of 1000 mg/kg bw, was orally administered, obviously because of the saturation of conjugation reactions. Other studies, showed that more than 95% of TBBPA administered orally is partially excreted as the parent compound and in feces it is eliminated in the form of metabolites within 3 days after a single dose with accompanying minute tissue retention or bioaccumulation, even at lower doses [11,12]. ln human plasma, the expected half-life of TBBPA-glucuronide is estimated to be between 2 and 3 days [13].

Additionally, the release and leaching of TBBPA residues into the environment have accelerated the problems of environmental pollution in recent times. Thus, it has adversely affected the ecosystem and caused their mobility in different matrices. The banning of hexabromocyclododecane (HBCD) and polybrominated diphenyl ethers (PBDE) has increased TBBPA production worldwide. This has led to their continuous existence in soil, food, consumer products, sediments, and dust, which indicates the continual threat they pose to humans by way of exposure, and may eventually bioaccumulate and biomagnify in organisms [14,15], subsequently causing negative health impacts due to its carcinogenicity and biotoxicity. Thus, the cessation of TBBPA utilization is still not sufficient to mitigate its potential adverse effects, after decades of continual exposure. For instance, TBBPA exposure could increase the risk of tumors, affect multiple cell types, endocrine disorders, neurotoxicity and reproductive toxicity [16,17] even at lower doses. However, these effects are poorly understood in humans, as investigations on their human health impacts is limited. Consequently, the goal of this paper is to examine the toxicity consequences of TBBPA, with an emphasis on how they influence the environment as well as human and animal health. Due to its increasing adverse health effects, investigations into its daily exposure to humans are urgently needed to assess its bioaccumulation and toxicity in the body. Fortunately, extensive investigations are being conducted to effectively detect and treat TBBPA.

For this review, we focused on the emerging methods for detecting TBBPA as well as strategies for treatment while discussing human biomonitoring, toxicity, and mechanism of action in the human body. Overall, research viewpoints that enhance the scientific understanding of TBBPA were covered, which will lead to breakthroughs for treating waste products containing TBBPA. The graphical abstract gives a detailed overview of the review.

## 2. Human Exposure Pathways and Bioaccumulation

### 2.1. Exposure by Diet Ingestion

According to [18], human exposure to TBBPA primarily occurs through ingestion. It has been demonstrated that TBBPA is present in animals and plants that humans consume. However, few studies have been carried out to detect TBBPA in food samples. For example, this chemical has been traceable in about 83% of food samples examined in the Fifth Chinese Total Diet Study and was above the maximum level. The mean concentrations in milk, aquatic foods, eggs, and meat were 5.76 ng/g lw, 3.05 ng/g lw, 3.12 ng/g lw, 1.78 ng/g lw, respectively [19]. Another study revealed that the daily dietary TBBPA intake increased from 0.256 to 1.34 ng/kg bw [20].

The results from food samples analyzed in South Korea for the presence of TBBPA revealed a higher concentration in sausage (0.049 ng/g ww) than in fish (0.021 ng/g ww) and cephalopods (0.023 ng/g ww). The average concentration found in sausage is a result of its high fat content. TBBPA is known to be highly lipophilic, which enhances its easy accumulation in fat. Moreover, the potential of TBBPA to accumulate in water [21,22], accounts for its concentration measured in cephalopods and fish [18]. These findings are alarming since, in addition to TBBPA being prevalent in the environment, their presence in food samples leads to human exposure, which has detrimental consequences. 

### 2.2. Exposure by Dust Ingestion and Inhalation

Several epidemiological studies have considered dust ingestion as the major exposure pathway to TBBPA. However, there are relatively few studies on TBBPA concentrations reported in air samples due to the high lipophilicity and low vapour pressure of this chemical compound. Moreover, its occurrence in office and household dust has partly been ascribed to its volatilization from consumer products that contain TBBPA. In Eastern China, a study was carried out in a printed circuit board (PCB) plant, and the concentration of TBBPA detected was 6.26–511 pg/m [23]. Higher levels (12.3–1640 pg/m) were recorded in indoor air from Shenzen, China, which contributed 4% exposure by inhalation and 76% by ingestion [24]. Interestingly, most researchers mentioned that lower levels of TBBPA have been recorded in outdoor air compared to indoor air. Barghi et al. [25] conducted a similar study in South Korea on 124 vacuum dust samples from kindergartens, schools, homes, cars, offices, public indoor areas and 32 samples from surface dust. Data from the study revealed that the TBBPA concentrations in indoor dust ranged from 78.87 to 463.81 ng/g, which was higher compared to reports from other countries. This could be as a result of Korea’s high demand for TBBPA. However, in living rooms, the concentration of surface dust was high at a significant value of *p* < 0.05, compared to bedrooms. Another report from Beijing, China revealed a higher TBBPA concentration (26.7 ng/g) in office dust, compared to the concentrations recorded in homes [26]. Wang’s group [27] also compared, dust samples collected from houses in different countries, the highest TBBPA concentration (2300 ng/g dw) was found in China. A report from a waste electronic recycling site showed that indoor dust had a higher concentration (3435 ng/g dw) of TBBPA than outdoor dust (1998 ng/g dw) [28]. Additionally, the group also studied levels of TBBPA in children and adults that lived close to the site, they found a concentration range of 0.31–58.54 ng/kg bw/day for children and 0.04–7.50 ng/kg bw/day for adults. A similar and broader study was carried out in twelve countries to quantify indoor dust for TBBPA. The estimated daily intake was ten times higher for children and infants in China, South Korea, and Japan than in other countries. Generally, the authors observed that the TBBPA intake was proportional to age, babies and children had a higher proportion (0.01–3.4 ng/kg bw/day; 0.01–1.2 ng/kg bw/day); however, the proportion decreased in adolescents and adults (0.003–0.61 ng/kg bw/day; 0.001–0.28 ng/kg bw/day), respectively [27]. Recently, Waiyarat’s group [29] investigated TBBPA concentrations quantified in house dust from rural and urban residential areas around an electronic-waste dismantling site in Thailand. A much higher concentration was recorded in houses around the e-waste dismantling site (median = 720 ng/g; range = 44–2300 ng/g) and a relatively lower concentration in urban (68.6 ng/g; 3.5–300 ng/g) and rural residential areas (17 ng/g; 2.0–201 ng/g) as a result of their distance from the dismantling site. The group estimated the daily intake of dust ingested and the results showed that the ingestion rate was relatively higher in toddlers than in children and adults. The authors concluded that activities from site could contribute to the contamination of house dust via TBBPA emissions. These results suggest that TBBPA has been detected in dust samples occurring as a result of household, occupational, and environmental emissions, which poses potential risk exposures from oral, dermal, and inhalation routes, particularly among children through hand-to-mouth contact [30]. According to the findings shown above, TBBPA accumulates more in indoor dust. This is a problem because most people spend a significant amount of time indoors, so they are more likely to be exposed to TBBPA and this could result in adverse health effects. Moreover, there is a higher susceptibility to TBBPA ingestion in babies and children than in adults as a result of crawling and hand-to-mouth contact. This is perturbing, due to their weaker defenses and body weight.

### 2.3. Exposure by Dermal Contact

Over the years, most studies have focused on the major pathways by which TBBPA is exposed through inhalation and ingestion. Considering dermal contact, Abdallah et al. [31] carried out an in vitro analysis on human skin, their study revealed that the absorption of TBBPA is dependent on a number of factors such as the time of contact and the dose administered. From the results, the range at which TBBPA was absorbed was 5.4 to 6.8%. Another in vitro study showed that TBBPA could penetrate the skin in a day at 3.5% [9]. The results of a different ex vivo study, showed that the human skin could absorb TBBPA at 53% and penetrate at 0.2% [9]. Yu’s group [32] dermally exposed Wistar rats to 600 mg/kg for 360 min daily for 3 months via an in vivo study, the relative absorption in male rats was 3–11% and female rats was 3–13%. However, upon exposure for 24 h, the relative absorption was 24.7% in male rats and 20.1% in female rats, respectively. Another in vivo study performed in rats revealed that the rate of penetration and absorption of TBBPA via the skin is 1 and 26%, respectively [9]. Despite these results, it has been proved that the dermal bioavailability of TBBPA is low compared to other brominated flame retardants. Knudsen et al. [33] exposed 100 nmol/cm^2^ of TBBPA for 24 h, and its rate of bioavailability was about 1 to 2%, which is equivalent to <1% for a high dose of ten-fold. With regards to the lower dermal absorption and penetration levels of TBBPA, it is necessary that researchers conduct more studies on the exposure pathway to the skin, so that there can be a better comprehension of this exposure pathway. 

### 2.4. Bioaccumulation and Excretion of TBBPA

Currently, studies exploring the accumulation of TBBPA in animals are few and most of them have utilized fish species. Findings from these reports have demonstrated that this substance is absorbed quickly and accumulates in a variety of aquatic organisms, such as zebrafish, bluegill sunfish, whelks and scallops [34,35,36]; their bioaccumulation rate is about 19.33%, while the rate of metabolism is 8.88% [37]. In rats, there is a similarity in the oral and dermal exposure to TBBPA with regards to their disposition and kinetic profiles [9,33]. Borghoff et al. [38] demonstrated in their study that the administration of a higher dose of TBBPA resulted in a higher concentration of the compound and its metabolites in the uterine tissue, plasma, and liver. Knudsen and co-workers [39] showed in their study that the fetus and offspring of Wistar rats could be affected after TBBPA is administrated to pregnant and nursing rats. Therefore, it is necessary to determine how the next generation will be affected after maternal exposure. Furthermore, TBBPA can be reduced via intestinal microflora to generate tribromobisphenol A, that can conjugate with glucuronic acid. After the TBBPA conjugates undergo enterohepatic recirculation, the metabolites are mainly excreted through the bile, urine, and feces. However, excretion through the bile in humans will cause the systemic bioavailability of TBBPA to decrease [8]. Research has shown that the main route for the excretion of TBBPA is via feces. Hakk’s group [40] demonstrated in their study that after 3 days of TBBPA administration in male rats a total of 92% was excreted in feces and 0.3% in urine. A similar study conducted, showed that after female rats were exposed to different TBBPA doses (25, 250, and 1000 mg/kg) for 3 days, >90 % was obtained in feces and 2% in urine [11]. 

Generally, from the studies discussed above, dust ingestion is the most important pathway for tetrabromobisphenol A exposure, however in children, dermal contact must be taken into consideration, because they spend most of their time on the floor, which results in a higher exposure of household dust. In addition, age is another factor to TBBPA exposure, since reports have shown that TBBPA intake in children and babies is higher than in adults and adolescents. Alternatively, in rats, there is a similarity in the kinetics and disposition of oral and dermal exposure, and TBBPA administered to expectant mothers and nursing rats, which is likely to affect future generations. Although, these research works are heading in the same direction, with comparable results and conclusions, there will be a need to conduct more studies to examine the absorption, metabolism, and excretion in humans and animals.

## 3. Human Biomonitoring

Human biomonitoring is a strategic component in evaluating human exposure to pollutants and their possible health effects. This approach can also be applied to examine if technological advancements could influence human exposure, estimate concentration patterns, enhance epidemiological research to assess health risks or identify susceptible individuals, and to analyze the effectiveness of regulatory measures. Recent research has revealed the biomonitoring of TBBPA in human samples such as breast milk, hair, serum, urine and other bodily fluids. Interestingly, the benefit of using urine as a noninvasive biomonitoring medium is the potential to obtain large volumes of samples, minimal ethical alarms, and the ability to monitor both genders and populations of all ages. Generally, in population investigations, the collection of urine spot samples is usually employed, because exposure levels are adjusted based on the specific gravity or creatinine level in urine to compensate for its dilution in spot samples [41]. Ho et al. [42] investigated TBBPA in human urine samples in Hong Kong. One hundred and forty (140) voluntary samples were obtained randomly for the study. In all the urine samples, a concentration range of 0.19–127.24 µg/g creatinine was quantified; however, >85% of TBBPA was measured in urine. Recently, the presence of TBBPA in urine samples from three cities in China (Chengdu, Nantong and Shehong) were determined. The concentrations of TBBPA were 0.0793, 0.633, and 1.15 μg/L in Shehong, Nantong, and Chengdu, respectively [43]. 

In population studies, blood (plasma or serum) or breast milk samples have been used. Usually, the maximum TBBPA concentrations in serum is below 0.8 μg/L, with fundamental characteristics less than 0.1 μg/L [44]. The lipid adjusted TBBPA blood concentrations have been reported by some biomonitoring studies [45]. Typically, concentrations in plasma/serum are regulated to the lipid content to compare with other plasma or serum samples and/or matrices. TBBPA was detected in human plasma samples by employing tandem mass spectrometry coupled to high performance liquid chromatography and gas chromatography and solid-phase extraction (SPE) [46]. This work directly utilized sulfuric acid on the polystyrene-divinylbenzene SPE column to decompose the plasma lipids and TBBPA was detected in a range of 4 to 200 pg/g in the plasma samples, with a detection limit of 0.8 pg/g. Subsequently, this technique was employed to analyze TBBPA in plasma from occupationally exposed persons. Tay’s group [47] collected serum samples (n = 61) from Norwegian cohorts to detect the presence of some emerging pollutants, TBBPA was detected in 36% of the samples with a concentration range of <0.28–74 ng/g lw. However, the geometric mean of TBBPA in the samples was 9.4 ng/g lw. These results were similar to Kim and Oh’s study [48], where serum samples of Korean mothers were collected from 2009 to 2010 with a mean level of 10.7 ng/g lw. Dallaire et al. [49] conducted a study on a pregnant woman’s serum in Canada. The result of the study showed that TBBPA was detected at a relatively lower concentration (0.3 ng/L), however subsequent studies on other human sera indicated a higher concentration level of ≥480 ng/L TBBPA. Interestingly, our study was unable to find any research that investigated the distribution of TBBPA in specific blood fractions. Considering this, we think that more research should be performed on this topic to help the biomonitoring community figure out the best way to report the concentration of TBBPA in the blood. Correspondingly, suitable biomonitoring equivalents must be applied in interpretating biomonitoring data of TBBPA (total TBBPA or parent only) in plasma samples.

Research is underway to find non-invasive biomonitoring matrices to overcome the constraints associated with invasive blood sampling. Human hair is considered one of the non-invasive matrices for biomonitoring due to the simplicity and affordability of sampling, ease of transport, storage, sample stability, the ability to reveal temporal exposure patterns through segmental analysis, and the availability of relevant data on exposure to a variety of environmental contaminants over short to long periods of time [50]. Human hair could be a promising medium for investigating lipophilic chemicals owing to its comparatively high lipid content (2–4%) [51]. Nevertheless, some drawbacks limit the ability of hair to be extensively utilized as a biomonitoring matrix for flame retardants. For example, only a small amount of hair samples can be obtained per person (normally 0.05–0.2 g), and additionally, individuals might not want to give away hair. Moreover, interpretating levels of pollutants in hair is not descriptive as they reflect endogenous exposure via blood contact at the follicle or hair roots and exogenous exposure through deposition from dust and air. Differentiating between these routes of exposure is challenging, particularly when hair is utilized as an indicator for pollutant exposure, while atmospheric deposition contributes to the level of exposure [52]. Barghi et al. [53] investigated TBBPA concentrations in human hair with nonspecific exposure obtained from 15 participants in Iran and 24 in Korea. The concentration of TBBPA ranged from N.D to 16.04 ng/g. TBBPA concentrations were significantly higher in Korean hair samples than in samples from Iran (*p* < 0.05). The authors also examined factors that could affect contributions of internal and external contaminations in hair. They concluded that the presence of TBBPA in the hair samples was most likely to be from internal exposure. Owing to the complexity of acquiring paired serum and hair samples, few correlation studies have been conducted on other flame retardants, but not TBBPA.

Considering the lipophilic nature of TBBPA, it tends to accumulate in matrices related to perinatal exposure, such as cord blood, breast milk, fetal membranes and placenta [54,55,56]. Although amniotic fluid and cord blood employed for human biomonitoring sampling is difficult, breast milk is usually used to monitor child and mother exposure to TBBPA [57,58]. In contrast to blood sampling, it is easier to obtain large quantities of breast milk without assistance from medical personnel. Moreover, breast milk samples only provide data on the exposure of specific populations, compared to blood samples. Shi et al. [59] reported a study on 24 breast milk samples collected in 12 provinces in China (not including Beijing) which were tested for the presence of TBBPA, and the mean level recorded was 0.93 ng/g lipid weight with detection limit of 5.12 ng/g lw. In Japan, a higher level of TBBPA (3.0 ng/g lw) was recorded in 64 breast milk samples [20]. According to another report, in Beijing, China, 103 breast milk samples were collected and detected for the presence of TBBPA; out of these collected samples, TBBPA was determined in 55 samples with limit of detection (LOD) less than 12.46 ng/g lw. The median and mean TBBPA concentrations were 0.10 and 0.41 ng/g lw, respectively [60]. Huang’s group [57] investigated the presence of TBBPA in 111 breast milk samples from 37 nursing mothers in Beijing. All the mothers donated one sample each month for three months. TBBPA was detected at a frequency of 64%, and the level of contamination was measured at a median level of 1.57 ng/g lw. The study also estimated the daily intake (EDI) of TBBPA through breastfeeding for nursing babies to be 6.62 ng/kg/bw/day. Inthavong et al. [61] collected 106 breast samples from French mothers and the TBBPA level detected was 0.5 ng/g lw. In Boston, USA, breast milk was sampled from nursing mothers, out of the percentage collected 35% of the samples contained TBBPA [62]. Kim and Oh [48], observed in their study that TBBPA concentrations were higher in babies than in their mothers. However, there was a dramatic decrease in the TBBPA concentrations with age for infants between 2 and 3 months after birth, which could be a result of short half-life, high maternal transfer, and a faster rate of excretion. Furthermore, significant correlations for TBBPA concentrations were found between babies and mothers, suggesting that maternal transfer was important.

Considering, the research carried out so far on human biomonitoring of TBBPA, it is obvious that a combination of different sources of exposure, could elucidate the variability in the high levels recorded in human samples. Therefore, it is important to assess levels of contamination in human tissues to provide evidence about the body intake and potential health risks from the general population to further investigate the exposure pathways and bioaccumulation of this hazardous contaminant, which will enhance a better understanding of the possible molecular toxicity mechanisms. The summary of the concentrations of TBBPA detected in human samples is shown in Table 1.

## 4. Toxicity Studies and Assessment of TBBPA on Humans

With regards to human health, there is still an open question about the possible health effects of TBBPA on the general population. Moreover, data on toxicological and human exposure indicate that exposure to this compound is detrimental to human health. Wu’s group [72] carried out a study on the effects of TBBPA exposure on human airway epithelial cells (A549) the analysis revealed that there was an increase in lipid peroxidation, ROS generation and caspase-3 activities. After an administration of 64 μg/mL TBBPA to the cells the mitochondria were severely injured and the smooth endoplasmic reticulum dilated. These effects could lead to the pathogenesis of a number of diseases. An investigation on the effects of TBBPA on human mesenchymal stem cells (hMSCs) revealed an increase in lipid droplets and upregulation of the expression of adipocyte-associated mRNA, alkaline phosphatase (LPL), and adipocyte-specific protein 2 (aP2) via a peroxisome proliferator-activated receptor gamma (PPARγ)-dependent mechanism. Considering the obtained data, the researchers concluded that lipogenesis in osteoblast differentiation was triggered by TBBPA, which could be reliant on increased expression of PPARγ [73]. Moreover, TBBPA upregulated apelin expression and secretion in epithelial ovarian cancer cell line (OVCAR-3), which is controlled by PPARγ [74]. Moreover, in granulosa tumor cells, the apelin receptor expression level was lower than in epithelial cancer cells, however, this was in contrast with the apelin expression and secretion [74]. TBBPA also disturbed the redox balance in human erythrocytes [75]. Again, Jarosiewicz’s group [76] confirmed in their study that TBBPA resulted in oxidative variations in human red blood cell membrane proteins and in human serum albumin proteins. This chemical also changed the conformation of albumin properties, which impaired α-helix and caused an increase in the content of the ß-sheet structure.

TBBPA has also proven to alter the tumor destroying function of NK lymphocytes, the secretion of IL-1β, and the inflammatory cytokines interferon gamma (IFNγ) [77,78]. It has been demonstrated that human immune cells such as NK cells and peripheral blood mononuclear cells exposed to TBBPA led to a decrease in TNF secretion, and ability of NK cells to bind to target cells [78]. Park et al. [79] observed that 10 mM TBBPA increased the expression of interleukin (IL)-8, IL-6, prostaglandin E2, and suppressed the release of the transforming growth factor beta 1 (TGF-β) in human first trimester. An investigation conducted on human bronchial epithelial cells exposed to TBBPA caused the expression of ICAM-1 and IL-6 to increase. Subsequently, the expression of proinflammatory proteins and signaling pathways of the nuclear receptor was disrupted [80]. Moreover, the effects of TBBPA at the vascular level is poorly understood. Hence, Feiteiro’s group [81] investigated the direct effects of TBBPA on the human umbilical artery (HUA) after 24 h exposure and examined its signaling pathway. The results obtained showed that the direct effects of TBBPA exposure on HUA induced vasorelaxation. They observed that the vasorelaxant response pattern of sodium nitroprusside and nifedipine was modified, which impaired the mechanism of the main HUA vasorelaxant. Interestingly, the genomic effects of this compound triggered an increase in vasodilation, this could be attributed to the connection of TBBPA with the NO, cGMP, sGC, PKG pathway. In addition, TBBPA modified the BKCa 1.1 α- and β1 -subunit channels, L-type Ca2+, sGC and PKG protein. Thus, after TBBPA exposure at the vascular level, TBBPA induced changes in HUA.

Cancerous diseases are potential health effects related to the exposure of TBBPA. Lyu and co-workers [82] demonstrated that TBBPA could induce migration and invasion in human hepatocellular liver carcinoma cell lines (HepG2) in a dose-dependent manner by altering their number and lysosome distribution. TBBPA caused a decrease in the levels of protein in Cathepsin B (CTSB), Cathepsin D (CTSD) and Beta-Hexosaminidase (HEXB) intracellularly, on the other hand CTSB and CTSD increased extracellularly. Hence, the authors concluded that in cancerous cells exposure to TBBPA could cause lysosomal exocytosis. Furthermore, in this study the results of molecular docking proposed that TBBPA may bind to transient receptor potential mucolipin-1 (TRPML1), and this could regulate the calcium-mediated lysosomal exocytosis, thereby boosting metastasis in cancerous liver cells [82].

In general, results from human health studies suggest that TBBPA could promote the pathogenesis of some human disorders and lipogenesis in osteoblast differentiation, via increased expression of PPARγ. Similarly in rats, TBBPA can affect osteoblast function via oxidative stress and mitochondrial dysfunction. In addition, in the immune system, this compound could interfere with the tumor killing function of human natural killer (NK) lymphocytes, the ability of NK cells to bind to the target and the secretion of inflammatory cytokines. In addition, TBBPA exposure could alter vascular homeostasis of the human umbilical artery. In cancerous cell lines, lysosomal exocytosis and metastasis in the liver could be promoted by TBBPA. Furthermore, this compound has demonstrated to bind to estrogen receptors, which is consistent with the effects observed in rats. Nevertheless, reports on this subject are few, thus research should be instigated to elucidate the possible health concerns of TBBPA on the overall population.

### 4.1. Toxicity of TBBPA on the Reproductive System

TBBPA greatly decreases the activity of sex hormones and worsens infertility in both men and women [83,84]. TBBPA toxicity can increase the concentration of hormones such as, estradiol, luteinizing hormone (LH) and progesterone, which can trigger a decrease in cortisol concentration in serum. According to Hales and Robaire [83] TBBPA can induce the synthesis of testosterone in cultured MA-10 Leydig cells and alter the expression of steroidogenic genes. Liang’s group [85] also concluded that TBBPA treatment could downregulate the viability of spermatogonia cells, which can affect different endpoints, such as agitation of the cytoskeleton, and an increase in apoptosis. In an in vitro study, embryonic stem cells (ESC), exposed to TBBPA resulted in cell death, an increase in the generation of reactive oxygen species and mitochondrial malfunction [86]. In pregnant women, exposure to different concentrations of TBBPA can cause miscarriage, affect the developing fetus and maternal health as well as premature birth [87,88,89]. Prenatal exposure could lead to a higher risk of anemia in the third trimester [90]. The reproductive ability in males can be reduced, as a result of increased concentrations of TBBPA, which can alter the mobility of sperm cells, sperm concentration, displace the lateral head, and angular displacement [91,92]. Loss of sexual desire, low sperm count and erectile disfunction was also noticed in males exposed to TBBPA [93,94].

### 4.2. Induced Toxicity of Developmental Systems

Reports have shown that maternal exposure to TBBPA adversely affected the neurological and mental development of infants. The early stages of heart development could be affected after increased exposure doses of TBBPA in humans [95]. During embryonic development it was observed that the cardiac and nervous/skeletal muscle systems were severely impacted by TBBPA [92,96]. In addition, there is insufficient proof that TBBPA causes autism spectrum disorders in children. However, a study conducted revealed higher concentrations of TBBPA in the plasma of autistic children. Boys particularly reacted and behaved strangely due to elevated TBBPA levels [97].

### 4.3. TBBPA Induced Metabolic Disorders

Disruption of neuroendocrine function results in the emergence of metabolic diseases and other related health conditions. TBBPA might interrupt metabolic and normal physiological activities leading to chronic diseases such as, neurodevelopmental disorders, immune system disorders, and endocrine disorders at lower concentrations. Research has shown that glucose levels in human hepatoma cells corresponded proportionally to the levels of TBBPA [98].

### 4.4. Other Health Conditions Related to TBBPA

TBBPA triggered significant metabolic alterations in human hepatoma cells [98]. Liver disease and liver damage are often caused by TBBPA. There is enough evidence to prove that respiratory problems like lung diseases and pulmonary fibrosis are triggered by TBBPA exposure, even though the epidemiology of TBBPA on the respiratory system lacks clarity [99]. The production of reactive oxygen species and osteoblast apoptosis can be activated by TBBPA, which can lead to diseases of the bones [100]. A graphical illustration of the toxic effects of TBBPA is shown in Figure 1.

## 5. Mechanism of Action of TBBPA

### 5.1. Effects of TBBPA on Immunosuppression and Inflammation

Current findings on the health implications of TBBPA have revealed the possibility of endocrine disruption and neurological toxicity [1,85,101]. Due to the fact that humans are exposed via inhalation of dust/air, the respiratory system could be susceptible to TBBPA. In addition, little is known about its impact on the immune and respiratory systems. Koike et al. [80] observed that exposure to TBBPA could not increase the expression of interleukin-8 (IL-8) but rather interleukin-6 (IL-6) and intercellular adhesion molecule-1(ICAM-1). TBBPA stimulated the epidermal growth factor (EGF) production and phosphorylation of the epidermal growth factor receptor (EGFR). Inhibitors of the mitogen-activated protein kinase and EGFR-selective tyrosine kinase obstructed the increasing expression of proinflammatory proteins. Additionally, they studied the modulation for nuclear receptors which revealed ligand activity for thyroid hormone antagonist and thyroid hormone receptor suppressed the increase of the expression of IL-6 and ICAM-1 significantly. They concluded that in bronchial epithelial cells the expression of proinflammatory proteins can be obstructed by TBBPA, through either the variation of nuclear receptors or EGFR-related pathways. Another study was conducted to examine whether TBBPA activates inflammatory pathways, particularly the production of prostaglandins and cytokines, in the first trimester placental cell line HTR-8/SVneo in humans. The lowest concentration of TBBPA enhanced the release of IL-8, IL-6, and prostaglandin E2 (PGE2). It also suppressed the release of TGF-β in HTR-8/SVneo cells [79]. 

Furthermore, human natural killer lymphocytes have the potential to damage tumor cells and cells that are virally infected. TBBPA exposure as an environmental hazard had a noticeable effect on NK cells’ tumor-destroying (lytic) function as well as their ability to bond to target cells, leading to the interference with NK cell lytic function [102]. TBBPA exposure reduced the production of tumor necrosis factor-alpha (TNF-α) in all immune cells, irrespective of their composition [78], and also influenced the lytic activity of the NK cell by reducing NK cells’ potential to bond to targeted cells [102]. 

TBBPA treatment significantly increased the expression of matrix metalloproteinase-9 (MMP-9) and its promoter activity in human breast cancer MCF-7 cells. Transient transfection with MMP-9 mutant promoter constructs established that TBBPA’s effects are mediated by nuclear factor-kappaB (NF-κB) and activator protein-1 (AP-1) response elements. Additionally, the expression of TBBPA-triggered MMP-9 was facilitated by activation of the AP-1 and NF-κB transcription owing to the phosphorylation of mitogen-activated protein kinase (MAPK) and Akt signaling pathways. Furthermore, a particular NADPH oxidase inhibitor and a ROS scavenger inhibited TBBPA-stimulated activation of the MAPK/Akt pathways and MMP-9 expression. Findings from this study indicate that TBBPA could promote cancer cell metastasis in MCF-7 cells via the release of MMP-9 via ROS-dependent MAPK and Akt pathways [103]. 

### 5.2. Oxidative Stress

After entering the body, TBBPA could hinder cell function and affect human health adversely by acting as a thyroid hormone. To date, our understanding on the possible health effects of TBBPA on humans is restricted to results of in vitro assays. ROS have been determined in a variety of human cells, including lung cancer cells, immune cells, normal cells and using in vitro assays. TBBPA was found to increase ROS production in each case. In addition, TBBPA induced ROS in human neutrophil granulocytes in a concentration-dependent manner [32]. Choi et al. [100] argued in their research that TBBPA also reduced ATP levels, resulting in necrosis or apoptosis as well as a decrease in the release of cyclophilin A and B. TBBPA also boosted malondialdehyde (MDA) levels in human airway epithelial cells (A549), ROS production, and caspase-3 activity. However, cells were found to have dilated smooth endoplasmic reticulum and severely damaged the mitochondria after high levels of TBBPA exposure [72]. The nuclear receptor peroxisome proliferator-activated receptor γ (PPAR γ) has demonstrated to be a key metabolic transcriptional controller, with roles in obesity, insulin resistance, inflammation, atherosclerosis, tumors, and control of glucose metabolism. TBBPA increased the expression of aP2, LPL, adipocyte-related mRNA in human mesenchymal stem cells via a PPARγ-dependent method. It also raised the amount of lipid droplets in the body [73]. According to Hoffmann et al. [74] TBBPA boosted epithelial ovarian cancer cell line (OVCAR-3) secretion and apelin production, which does not involve estrogen receptors but the peroxisome proliferator-activated receptor γ. Furthermore, the expression of apelin receptor was higher in epithelial cancer cells than in granulosa tumor cells, although apelin expression and secretion were the contrary.

### 5.3. TBBPA Induced Genotoxicity

In human peripheral blood mononuclear cells, a lower concentration of TBBPA induced DNA damage, stimulated oxidative mutilation to pyrimidines and purines [104]. A low TBBPA concentration was known to down-regulate the transmembrane transport of the cell membrane-related genes and cellular skeleton, proving that the membrane damage was genetically controlled. It has been reported that the expression of profibrotic genes and proteins were upregulated after a short-term TBBPA exposure [105].

### 5.4. TBBPA Induced Neurotoxicity Dysfunction

TBBPA exposure can cause neurotoxicity and interrupt the function, activity and production of some hormones (thyroid, estrogen) as well as MAPK, PPAR-related signaling pathways. These pathways control transporters of the central nervous system (CNS) barriers. Cannon and co-workers [106] observed in their study that TBBPA induced permeability variations in the blood brain barrier, by altering brain homeostasis, obstructing CNS drug delivery, and increasing the brain’s exposure to this chemical compound. Another study showed that TBBPA affected the development of neural ectoderm, which impacted neuron transmission, axon growth and dysregulated the signaling pathways of wingless-related integration site (WNT) and aryl hydrocarbon receptor (AHR) in human embryonic stem cells [98]. TBBPA can potentially affect the expression of human neural stem cell identity and neurogenesis by competing with NOTCH, GSK3β, and T3 signaling [107]. Levels of the presynaptic protein SNAP-25 could be increased by TBBPA exposure. This plays a significant role in exocytosis and intracellular vesicular transport, which is related to cognitive abilities and hyperactivity in some neuropsychiatric conditions [108].

## 6. Environmental Impact of TBBPA

Several studies have focused on the environmental fate of TBBPA; thus, it is necessary to review the data from literature to comprehend the presence of this chemical in the environment as well as its biological effects on living organisms [109]. TBBPA has the potential to cause ecological and bio-health issues because it is highly volatile, lipophilic (log Kow = 4.5), with low solubility in water (0.72 mg/mL) and possess bio-accumulative characteristics. Considering its widespread production, environmental encounters are unavoidable. Additionally, the behavior and distribution of this compound in different biotic systems and environments can be attributed to its chemical properties [110]. For instance, the solubility of TBBPA increases when pH increases. TBBPA has a higher solubility than other BFRs such HBCD and PBDE [111,112] even at neutral pH. As an additive flame retardant (AFR), this compound could be released from consumer products and get into dust and indoor air [113]. When TBBPA is utilized as an AFR, it is applied mainly in the exterior casing of electronic gadgets/devices, for example in computers and televisions. Some studies have shown that the main TBBPA sources in dust and indoor air were from old electrical appliances and electronic products, especially televisions, computers, etc. [114]. For studies conducted on dust samples, a concentration of 2660 ng/g dw was found in the printed circuit board manufacturing plant mentioned above in Eastern China [23]. Therefore, areas with a high population density, and a strong activity region, which has a large-scale application of electronics containing TBBPA appears to be a significant contributor to its increased concentration in the environment. In addition, studies conducted showed TBBPA was detected in leachate collected from eight landfill sites in South Africa. The data obtained showed that the TBBPA found was below the detection limit [115]. The concentration of TBBPA found in housing plastics collected from two e-waste recycling companies in China was 0–34 mg/kg [116]. Moreover, higher TBBPA concentrations (24.69–913.64 ng/g dw) have been detected in the Longtang River tributary in Guangdong, China, which is closer to a known disposal and e-waste site [117].

Worldwide monitoring data indicates that TBBPA is detected at low levels in the water, air, soil, and sediment in distant and urban localities. However, higher concentrations can persist in sediments and soils close to the manufacturing facilities and e-waste recycling facilities of brominated flame retardants. TBBPA-based products are not the only sources of TBBPA in the aquatic environment but also through effluents from landfill sites and wastewater treatment plants. TBBPA is harmful to a wide range of aquatic organisms and can threaten their reproduction, development, and survival even at relatively lower concentrations [21]. Nevertheless, in China, TBBPA concentrations in the aquatic environment were compiled, and the results were substantially below the limits, as the Chinese criterion maximum concentration (CMC) is 0.1475 mg/L and the criterion continuous concentration (CCC) is 0.0126 mg/L for aquatic exposure [118]. Gong’s group [21] detected TBBPA in surface water at a concentration range of ND-0.46 μg/L ww. In another study, a group of researchers collected 36 water samples from tributaries and the main stream of the WeiHe River Basin. The TBBPA concentration ranged from ND-12.279 ng/L with a mean value of 0.937 ng/L [119]. Again, TBBPA was found in water samples collected from Fuhe river and Baiyang Lake. After analysis the concentration range was from 18.5 to 82.6 ng/L [120].

Research has revealed varying concentrations of TBBPA in sediments. Wang et al. assessed TBBPA concentration in sediments to be N.D-3.889 ng/g with a mean value of 0.283 ng/g in the WeiHe river [119]. Elevated levels has been found by Yang’s group [34] in sediments, which ranged from 19.8 to 15.2 µg/g dw. A group of researchers collected sediment samples from the Xiaoqing River and Lianjiang River to examine the occurrence of TBBPA. Data from the study showed TBBPA was found in both rivers. The concentration found in the Xiaoqing river were 0.76–2.51 ng/g dw and 108–31 µg/g dw in the Lianjiang river, respectively [121]. Similarly, sediments were assessed for the presence of TBBPA in two rivers in South China. The level found was in the range 0.003–0.31 ng/g dw [122]. The study of Pan et al. [123] revealed that sediment samples collected in fishing ports along the coast of South China contained TBBPA in the range 0.02–21.5 ng/g dw. 

There have been a few reports on the investigation of TBBPA in biota samples. Biota samples collected from the Pearl and Jiulong river estuary had TBBPA concentrations from 0.56 to 22.1 ng/g lw [122]. In another study, the level of TBBPA measured in biota was 696 to 197 µg/g lw [34]. It was revealed that in the Laurentian Great Lakes of North America, herring gull egg pools and specific eggs from 14 colony sites contained TBBPA. The researchers observed higher concentrations (<LOD- 42.8 ng/g ww) in collected eggs from pools, whiles for specific eggs, the concentration ranged from <LOD to 497 ng/g ww [124]. 

The soil is predicted to be the primary TBBPA contamination matrix. In Chongqing, China, levels of TBBPA contamination (<LOD to 33.8 ng/g dw) have been recorded in soils around a residential area [125]. Moreover, TBBPA concentrations altered in soils from farmlands, dismantling sites, residential areas, and industrial areas. Jeon and co-workers [126], also detected TBBPA (4.4 ng/g) in soils from farmlands in South Korea, the TBBPA concentration was between 29.98 and 165.79 ng/g in South China, which is higher than that recorded (0.025–78.6 ng/g) in East China from an industrial site [127]. However, TBBPA concentrations in soils were higher in China, compared to concentrations recorded in South Korea. 

TBBPA may be released from wastewater treatment plants (WWTP) into the environment through landfilling of sludge and effluent discharge to surface water [48,128]. The environmental loading from sludge and effluent depends on the existence of TBBPA in influent wastewater, volume of treated wastewater, and the removal efficiency of WWTP. When a high removal efficiency is recorded, a great amount of the chemical may be found in the sewage sludge [129]. Song’s group [130] investigated 52 sludge samples collected from thirty WWTP in China and concentration ranged from 0.4 to 259 ng/g dw. Similar reports were observed in Spain (N.D to 472 ng/g dw) [131], and South Africa (38.58 ng/g) [132]. Zhu et al. [133] detected relatively lower TBBPA concentrations (1.36–195 ng/g dw) in sludge from WWTP, which exceeded the maximum levels described in South Korea (4.01–618 ng/g dw) [134]. Wastewater from industries manufacturing and utilizing TBBPA are projected to contribute large amounts of TBBPA in WWTP. Typical manufacturing centers may increase the occurrence of TBBPA in sludge at WWTP and industrial wastewater.

In Shanghai, TBBPA was investigated in a printed circuit board manufacturing plant, reports from the study showed that TBBPA in wastewater was in a range of 28.3 ± 9.19 to 174 ± 31.3 ng/L, treated effluent was in a range of 3.21 ± 0.50 ng/L, and dewatered sludge was in a range of 100 ± 4 ng/g dw [23]. Considering, the sludge levels reported in other countries, the results were within the category of other WWTP in China, this notwithstanding the data is insufficient to establish occurrence trends of TBBPA in waste printed circuit boards. Table 2 summarizes TBBPA concentrations in different environmental components and Figure 2 illustrates published works on TBBPA detected in environmental samples. These studies have shown that TBBPA is a universal contaminant. In addition, the environmental impacts of TBBPA are illustrated in Figure 3.

## 7. Emerging Methods for Tetrabromobisphenol A Detection and Treatment in the Environment

Tetrabromobisphenol A is mainly employed in epoxy resin to manufacture printed circuit boards, upholstery and textiles, carpet, electronics, children’s toys, mattresses, building materials and plastic products. Interestingly, the aromatic structure of TBBPA, is one of its significant chemical properties. Additionally, this compound, TBBPA, is known to act as an endocrine disruptor, a characteristic that enables it to interfere with normal hormonal functions and levels when it binds to target tissues in the human body. Therefore, several sample pretreatment procedures and analytical techniques have been utilized for TBBPA detection.

### 7.1. Sample Pretreatment

The complexity of environmental and food samples requires the need for pretreatment of samples prior to analysis. Sample pretreatment is arduous, which involves downstream methods and extraction for enhancing the reliability of the protocol. Hence, there is a necessity to keep improving existing methods and developing innovative methods.

#### 7.1.1. Liquid-Liquid Extraction (LLE) Method

The liquid-liquid extraction method has been extensively applied sample treatment owing to its robustness, low-cost and user-friendly operations. Despite its numerous advantages, it is not environmentally friendly due to the extensive usage of solvents, such as butane, hexane, acetonitrile for extraction. Seafood samples were analyzed for the presence of TBBPA and BPA using an improved quick, easy, cheap, effective, rugged, and safe (QuEChERS) procedure Acetonitrile and inorganic salts (NaCl, MgSO_4_) were added to efficiently extract the analytes followed by LLE with tertbutylmethyl ether/hexane and benzene/hexane. The limit of detection for TBBPA was 0.006 ng/g ww [141]. The organic phase of this method resulting from the clean-up stage was easily concentrated, which enhanced the sensitivity.

Few methods such as dispersive liquid-liquid microextraction (DL-LME) and molecular-complex-based liquid-liquid microextraction (MC-LLME) are newly designed green extraction approaches. DL-LME was employed to extract TBBPA from dust with a low-density solvent. The developed method had a good recovery rate (88.9%) [142]. HPLC-MS/MS coupled with electrospray ionization tandem mass spectrometry was employed to detect TBBPA in complex environmental samples, the limit of detection obtained in anaerobic sludge was 2 ng/g [143]. MC-LLME was utilized to analyze TBBPA and bisphenol A (BPA) in water samples with no dispersants [144].

#### 7.1.2. Microwave-Assisted Extraction (MAE)

Employing microwave energy for the extraction and pre-concentration of TBBPA constitutes a significant phase in MAE. MAE coupled with LLE has advantages such as less extraction time (20 min), less sample volume, less solvent consumption and sensitive to detect TBBPA with a low detection limit of 1.4 ng/g [145]. MAE followed by HPLC-MS was considered to be an adequate approach to extract TBBPA from plastic waste compared to ultrasonic-assisted extraction (UAE). Higher amounts TBBPA was recovered by using isopropanol/hexane [146]. Similarly, MAE was utilized to remove TBBPA from plastic waste within 30 min using hexane/Isopropanol and hexane/ethanol with recovery rate greater than 73% [147]. The introduction of ultrasound and microwave energy into QuEChERS technique increased the efficiency of extracting bisphenols including TBBPA significantly in serum and sediment samples [148]. BPA and TBBPA were detected in commercial milk samples by a rapid and facile microwave assisted ionic liquid microextraction (MAILME) method. This technique had a remarkable recovery range between 78.2 and 99.8% with LOD (0.02 µg/L) for TBBPA, and performance time of 5 min; interestingly, this approach did not require the removal of proteins and fats in the milk prior to analysis and no organic solvents were utilized [149]. Therefore, MAE has demonstrated to be a competent technique with superior extraction capacity, recovery rate and less solvent consumption thus reducing the pollution load in food and environmental samples.

#### 7.1.3. Solid Phase Extraction (SPE)

The SPE method is the most recommended technique in terms of efficiency, flexible operation, and TBBPA detection. A magnetic SPE was employed to extract 4-nonylphenol and TBBPA from water. This approach was robust and practical with LOD of 0.011 µg/L and recorded a satisfactory recovery range from 93.2 to 101.1% for TBBPA [150]. A swift solid phase extraction method coupled with ultra-high-performance liquid chromatography-tandem mass spectrometry (SPE-UPLC-MS/MS) was applied to sensitively investigate flame retardants, including TBBPA in water. The water samples were acidified and extracted via hydrophilic-lipophilic-balance solid phase extraction (HLB-SPE) cartridges. This technique had a recovery of 76.2–98.1% with low detection limit of 0.10 ng/L [151]. Similarly, another investigation was carried out in water to determine three brominated flame retardants, including TBBPA, using dispersive solid-phase extraction (DSPE) based on novel molybdenum disulfide (MoS_2_) modified with carbon dot (CD) coupled with HPLC. This promising method had a recovery of 80–91% and LOD in the range 0.01–0.06 µg/L [152]. Micro-nana-structured magnetite particles (MNMPs) were prepared and the surface was modified with citric acid molecules. The MNMPs served as a good adsorbent to extract TBBPA and HBCD from water with a recovery range of 83.5–107.1% [153]. TBBPA and HBCD in water samples were extracted with direct immersion solid phase microextraction (DI-SPME), this method exhibited a better performance time of 35 min, longer lifespan and usage of few organic solvents compared to accelerated solvent extraction (ASE) [154], liquid-liquid extraction (LLE), and SPE. Porous covalent organic framework (P-COF) was synthesized and applied for SPME coupled with constant flow desorption ionization-mass spectrometry (CFDI-MS) extract TBBPA analogs in sea and river water. The pretreatment time was short for each sample (7 min), P-COFs improved mass transfer and made the active sites of TBBPA analogs easily accessible, which resulted in a higher extraction efficiency (2.3–3.6) than commercial coatings and unadulterated microporous covalent organic frameworks [155].

#### 7.1.4. Molecularly Imprinted Solid-Phase Extraction (MIP-SPE)

Molecularly imprinted polymer (MIP) or engineered polymers are developed to adsorb and selectively detect compounds. TBBPA content in water was determined by using three templates diphenolic acid-MIP (DPA-MIP), bisphenol A-MIP (BPA-MIP) and non-imprinted polymers (NIP). The selectivity of DPA-MIP and BPA-MIPs were higher for TBBPA than NIP, when they were applied as sorbents for SPE. Comparatively, DPA-MIP had a superior adsorption capacity to TBBPA at low concentration levels than BPA-MIP, this could be as a result of the higher affinity binding sites of DPA-MIP for TBBPA, which enhanced the selectivity of the polymers with a recovery rate of 85–97% [156]. MIP with outstanding selectivity was employed for SPE. This methodology was applied successfully to extract TBBPA and other BFRs in sewage and sludge samples [157].

A combination of magnetic materials with molecularly imprinted polymers were applied for sample preparation. Magnetic molecularly imprinted polymer (MMIP) was developed further to form Fe_3_O_4_@SiO_2_@MIPs and coupled with HPLC for effective TBBPA detection in water samples. 4-Vinyl-pyridine and ethylene glycol dimethacrylate was utilized as the monomer for the TBBPA-MIP material [36]. Similarly, N-doped carbon-encapsulated iron surface molecular imprinting polymer was applied as an adsorbent and catalyst to determine TBBPA [158]. Likewise, Fe_3_O_4_-mTiO_2_-MIP was fabricated to extract TBBPA from polluted water. The designed MIP exhibited good repeatability because of its magnetic core [159]. Among the various techniques such as: LLE, MAE, LPME, SPME, SPE and MIP-SPE, the application of SPE/SPME has been considered to be a green and efficient pretreatment method for TBBPA analysis in complex samples. The utilization of MIP as a selective material has improved the separation efficiency. Moreover, automated sampling, simple operation, outstanding reproducibility, selectivity and accuracy, solvent-free analysis, application in different fields, are some of the benefits of SPME/SPE approach. However, LLE, MAE, LPME have shorter pretreatment time, simple operation, and cost-effective instrumentation. However, disadvantages such as purification steps, low precision, and higher usage of solvents deter their application for various field investigations [160]. Table 3 shows a summary of the sample pretreatment methods for TBBPA. 

### 7.2. Analytical Techniques for Detecting TBBPA Environmental Samples

TBBPA is typically found in the environment at lower concentrations, therefore analytical methods with high selectivity and sensitivity are needed for its detection. Over the years, different techniques have been employed to detect TBBPA in a quantitative manner in different matrices. At present, methods such as, gas chromatography mass spectrometry (GC-MS) [161,162,163,164,165], ultra-performance liquid chromatography (UPLC) [151,166], high performance liquid chromatography (HPLC) [167,168,169], and liquid chromatography tandem mass spectrometry (LC/MS-MS) [161] have being employed to detect TBBPA. Even though these methods have the advantage of high sensitivity in terms of identification and quantification, they have certain constraints, such as (i) complex and costly instrumentation, which requires skilled personnel to operate; (ii) they do not support on-site studies; (iii) low throughput, which cannot meet the requirement for samples analyzed in large quantities; and (iv) sophisticated pretreatment processes to sort the analyte from the sample. However, in recent years, immunochemical techniques such as fluoroimmunoassays [170] and enzyme linked immunosorbent assay (ELISA) [171,172] have been utilized as alternative detection methods for TBBPA. They have proven to be rapid, simple, reliable, and sensitive [173,174,175]. Furthermore, they have demonstrated the ability to analyze a large number of samples simultaneously in a short time. In addition to these high-throughput immunoassays, other effective analytical techniques have been designed, including colorimetry [176], molecular imprinting polymer [159], photoelectrochemical [177], and electrochemical sensors [107], which employ devices that are miniaturized, portable, robust and support on-site analysis. All of these methods have been applied to detect TBBPA, and they all have the same features, e.g., simple sample preparation, automation, and feasibility.

#### Detection Methods for TBBPA in Environmental Samples

Han et al. [103] analyzed TBBPA and other brominated flame retardants in different water samples (ground water, surface water, raw water, bottled water, tap water, and finished water samples). They employed solid-phase extraction-ultra-high-performance liquid chromatography-tandem mass spectrometry (SPE-UPLC-MS/MS) for the analysis. The target compounds were grouped as group 1 and group 2, because of the differences in their ionization modes and elution conditions. TBBPA belonged to group 1, all the samples were analyzed with UPLC-MS/MS, after they were run twice under different conditions. The LOD and linearity range were obtained at 0.1–2.5 ng/L and 0.1–100.0 ng/L, respectively. Firstly, the pH of the samples was adjusted between 2 and 3. Afterwards, hydrophilic-lipophilic-balance solid phase extraction (HLB-SPE) cartridges were employed to extract the acidified water samples eluted with acetonitrile (12 mL). The relative standard deviations and recoveries were in the range 2.0 to 28.5% and 76.2 to 98.1%, respectively. Based on electrochemical immunoassay Yakubu et al. [178] reported on the detection of TBBPA in water using successfully synthesized gold palladium bimetallic nanoparticles modified on amine (-NH_2_) functionalized-nanoflower-like manganese oxide (NH_2_-fMnO_2_). The NH_2_-fMnO_2_ was utilized as the signal tag and a carrier to coat the secondary antibody (Ab_2_), which reduced in H_2_O_2_ due to its peroxidase mimicking performance. Moreover, the electrode was modified with multi-walled carbon nanotubes (MWCNTs) to enhance the immobilization of TBBPA-antigens (Ag) and also created a large surface area for the antigen-antibody reactions. TBBPA oxidation was boosted by AuPd nanoparticles (NPs), thus improved signals were generated. Under optimum conditions, the immunosensor had a detection limit of 0.10 ng/mL with satisfactory accuracy (recoveries, 84 to 120). Another method which employed poly (sulfosalicylic acid PSSA) composite film and gold nanoparticles (AuNPs-PSSA) was developed by Shen’s group to detect TBBPA in wastewater. They fabricated AuNPs-PSSA on the surface of an electrode in-situ through cyclic voltammetry scanning [179]. Electrochemical analysis was employed to reveal the performance of PSSA film, AuNPs and AuNPs-PSSA. The results showed that a larger active surface area was observed for the composite film AuNPs-PSSA, its charge transfer resistance was lower, and its accumulation efficiency toward TBBPA was higher than the PSSA film and AuNPs. Hence, the sensing sensitivity of TBBPA and oxidation signals were enhanced on the surface of the AuNPs-PSSA significantly. The constructed platform for sensing TBBPA had an LOD of 25 pM and a linear range from 0.1 to 10 nM. In summary, the utilization of the electrochemical method to detect TBBPA has proved to be very successful in the sensing field, due to its rapidity and sensitivity. Hence, different emerging pollutants have been targeted for environmental monitoring using the electrochemical technology. 

Yang and colleagues [180] utilized ultra-high performance liquid chromatography-tandem mass spectrometry (UPLC-MS/MS) to detect tetrabromobisphenol A in soil samples. TBBPA was purified by employing acetone (1 mL) with LC-Si cartridge and dichloromethane/n-hexane (30 mL, 1:1, *v*/*v*) as the eluent after ultrasonic extraction. Electrospray ionization (ESI) was used to measure TBBPA in the multiple reaction monitoring (MRM) mode and atmospheric pressure chemical ionization (APCI) was employed to quantify their derivatives. Matrix and blank spiking experiments were used to assess the developed detection method. All the samples had a recovery rate in the range 78–124%. In the soil samples, method quantitative limits (MQL) of TBBPA were in a range of 0.22 to 8.8 pg/g dw. Furthermore, Fu et al. [181] developed another ELISA technique, to determine TBBPA in soil using fluorescence enzyme linked immunosorbent assay (FELISA). The TBBPA specific nanobody (VHH) obtained from camelid was modified with alkaline phosphatase to create a bi-functional fusion protein that can bind with TBBPA and produce a detection signal at the same time. The maximum half inhibition concentration (IC_50_), LOD, and linear range of the immunoassay were 0.23 ng/g, 0.05 ng/g and 0.1 to 0.55 ng/g, respectively. Owing to the higher resistance of organic solvents to the fusion protein, 40% dimethyl sulfoxide (DMSO) was utilized as the extract solvent to remove the matrix effect via a simple pretreatment, which resulted in good recoveries (93.4 to 112.4%) for the spiked samples.

The enzyme-linked immunosorbent assay has been utilized as an alternative approach because of its simplicity, low cost and ability to detect TBBPA with high throughput in different matrices. He and co-workers [182], detected TBBPA residues in sewage, sludge, and landfill leachate by developing an ELISA method which employed anti-tetrabromobisphenol-A nanobodies (Nb) biotinylated in three different valences (Nb1, Nb2, Nb3) and entrapped onto streptavidin (SA)-bacterial magnetic particles (BMP), which was utilized as a carrier substrate to develop Nb1-BMP-Biotin-SA, Nb2-BMP-Biotin-SA, and Nb3-BMP-Biotin-SA complexes. These complexes exhibited resistance to high temperature (90 °C), pH (12), methanol (100%), and strong ionic strength (1.37 M NaCl). Therefore, an ELISA based BMP-SA-Biotin-Nb3 was developed, which represented IC50 at 0.42 ng/mL. The range of TBBPA detected by the assay were <LOD-3.65 ng/g (dw), <LOD-0.75 ng/mL, <LOD-1.17 ng/mL, for sludge, sewage and landfill leachate samples, respectively. A similar method which employed a nanobody based immunoassay to detect TBBPA in sediment was successfully developed by Li’s group [181]. Basically, anti-TBBPA Nb was incorporated with nanoluciferase, to develop a highly sensitive one-step bioluminescent enzyme immunoassay (BLEIA), with IC_50_ of 187 pg/mL. Even though the developed assay was 10 times higher compared to the two-step classical ELISA, with IC_50_ of 1778 pg/mL. Detecting trace TBBPA in sediment is a still a challenge due to its relatively higher matrix effect. Therefore, the performance of the one-step BLEIA was enhanced, by inserting C4b-binding protein (C4BP) as a self-assembling ligand between the nanoluciferase and nanobody. Consequently, a heptamer fusion comprising of seven tracers and seven binders was produced, which enhanced the signal amplification and binding capacity. Interestingly, the BLEIA and one-step heptamer displayed an improved sensitivity of additional 7-fold (IC_50_ at 28.9 pg/mL) and a lower detection limit determined at 2.5 pg/mL. The developed immunoassays employed in the ELISA methods discussed above made tremendous improvements with regards to stability and sensitivity compared to other designed assays. Previously, most immunoassays utilized traditional antibodies to detect TBBPA. These assays were not sensitive enough due to the larger molecular size of the antibodies for TBBPA detection, which could affect the sensitivity. However, the studies mentioned above used nanobodies, which have a molecular size of about one-tenth that of traditional antibodies. Hence, these established immunoassays have created a breakthrough for the detection of other small molecules using nanobodies as they have prospects in immunoassay development.

Even though chromatographic methods are highly selective and sensitive they have drawbacks involving bulky and special equipment, require skilled technicians, time consumption, high cost, strenuous sample preparation and treatment, and inability to detect pollutants on-site, which limits their extensive application. In view of that, Zeng et al. [183] reported an LOD of 3 pg/g, with the range of the spiked recoveries from 94.4 to 106.6% and relative standard deviation values below 8.6% for the colorimetric detection of TBBPA in dust samples. Molecularly imprinted polymers prepared via the sol-gel method were used to recognize TBBPA selectively on a paper support consisting of a copper-based metal organic framework (MIP/HKUST-1). The maximum adsorption capacity was 187.3 mg/g, whiles the imprinting factor was 7.6, which is better than other developed MIPs. On account of the selective recognition by the MIP, the enzyme-like abilities of HKUST-1 under the MIP layer became weak as a result of the decreased imprinted cavities of the residue leading to the adsorbed TBBPA being degraded in the presence of hydrogen peroxide (H_2_O_2_). The synergy between HKUST-1 and H_2_O_2_ led to the color produced via catalytic oxidation of 3,3′,5,5′-tetramethylbenzidine to become less distinct, the gray intensity was proportional to the logarithm of TBBPA concentration, which was in the range 0.01 to 10 ng/g. The colorimetric method is a simple analytical approach, which is highly advantageous due to its rapid response, excellent accuracy, cost effectiveness, naked-eye readout, and ease of experimental operation [184,185,186,187,188]. This method has been effectively implemented in food analysis, environmental monitoring, medical diagnosis, and chemical sensing for the determination of different analytes. 

Photoelectrochemical (PEC) bioanalysis is a recently developed technique that has become the focus of innovative scientific research interests due to its outstanding sensitivity, simplicity, inherent miniaturization, and portability [189,190,191]. With advancements in material science and technology, a number of novel PEC approaches have been developed for the detection of pollutants in the environment [192,193,194,195,196]. However, the PEC method for TBBPA detection has been rarely investigated, Li and team members simultaneously detected TBBPA in water and indoor dust from an electronic-waste recycling site using a novel molecularly imprinted photoelectrochemical sensor (MIPES), which displayed a high sensitivity and rapid response [197]. The MIPES’ sensing ability toward the detection of TBBPA was assessed by the response of the photocurrent. The amperometry working potential was 0.48 V and photoelectrochemical detection wavelength range was 320–780 nm. The novel sensor had detection limit of 0.51 nM and a detection range of 1.68 to 100 nM as well as a limit of quantification of 1.68 nM. In addition, the relative standard deviation was below 7.0%, and the recovery range was between 90.0 and 115%. 

Recently, Guo and co-workers [198], developed an electrochemical sensor to selectively detect TBBPA in beverages based on magnetic carbon dots (MCD) and cetyltrimethylammonium bromide (CTAB) modified GCE. The synthesized MCDs exhibited a remarkable electrocatalytic activity, and the strong hydrophobic interface of CTAB boosted the enrichment capacity of hydrophobic compounds, and their combination enhanced the electrochemical signal further. Under optimum conditions, the linear range of constructed sensor was 1 nM–1000 nM, and a satisfactory detection of limit of 0.75 nM was achieved. Zhu et al. [199] considered the advantages of fluorescence methods such as simplicity, fast synthesis, sensitivity, selectivity, low cost, and ability to determine concentrations at trace levels to develop a facile competitive ratiometric fluorescent immunoassay to monitor TBBPA in environmental and food samples based on fluorescein amidite (FAM)-DNA–functionalized CdSe/ZnS quantum dots (QD). In the detection system, the secondary antibody (Ab_2_) was coated on catalase (CAT), which acted as the regulator of the concentration of H_2_O_2_. After the competition phase, FAM-DNA-functionalized CdSe/ZnS QDs emitted a red fluorescence at an excitation of 490 nm, which was quenched effectively by the added H_2_O_2_. Under optimal conditions, LOD was calculated as 0.118 μg/L with a linear range from 0.34 to 45.34 μg/L. In addition, the researchers combined dual-output ratiometric fluorescence assays with ELISA to increase the in-built rectification to the environment, resulting in acceptable accuracy (recoveries, 83.16 to 112.4%) and precision (2.42 to 7.28%).

To further improve the sensitivity of the previous method, Zhang’s team [200] proposed an interesting electrochemical sensor, which was integrated with molecularly imprinted polymers to monitor TBBPA in plastic products. The sensing platform was modified with reduced graphene oxide-silver nanodendrites (rGO/AgND) composites to improve the electrode performance. The selectivity of the modified electrode was enhanced by labelling it with MIP via electro-polymerization. In their work, the performance of the electrode was significantly improved because two modified layers were utilized, which were AgNDs and rGO. Additionally, they also established that certain factors have an effect on the developed sensor such as the proportion of ACN to water and the kind of eluent and elution method employed. Moreover, two outstanding linear relationships were exhibited by TBBPA concentration with peak currents between 0.05 nM and 20 nM with a limit of detection 0.015 nM. The TBBPA content successfully detected in phone cases and plastic water bottles were 56.00 and 1.484 μg/g, respectively. A fluorescent probe was designed by employing MIPs combined with CdTe quantum dot hybrid particles and wrinkled silica nanoparticles (WSN) to detect TBBPA in plastic electronic waste samples [201]. The adsorption capacity of the MIP-capped wrinkled silica-QD hybrid particles (WSN-QD-MIP) was 96.5 mg/g with an imprinting factor of 7.9 towards TBBPA. Under optimal experimental conditions, the fluorescence intensity (λ_ex_ = 340 nm, λ_em_ = 605 nm) was quenched, which was proportional to the TBBPA added. The LOD was 5.4 nM and the detection range was from 0.025 to 5 μM. The obtained results were in agreement with high-performance liquid chromatography. A fluorescent sensor was constructed based on molecularly imprinted polymer labelled on CdTe quantum dots (MIP-QD) to detect TBBPA in electronic-waste samples. The MIP-QD was characterized and it displayed unique morphological and photoluminescence properties. The designed fluorescent sensor had a concentration range between 1 and 60 ng/mL and a detection limit of 3.6 ng/g. The relative standard deviation was below 6.2% and recoveries ranged from 89.6 to 107.9%. The average TBBPA concentration detected in the circuit boards was 707.3 and 260.2 mg/kg for electric fan samples. A group of researchers used the property of fluorescence enhancement of MOFs to detect TBBPA for the first time [202]. In their work, a porous MOF-74(Zn) (Zn_2_(DHBDC)(DMF)(H_2_O)_2_, framework with open metal sites was prepared. To achieve a highly sensitive and selective fluorescence detector, MOF-74(Zn) was imbedded on the open metal sites with ethylenediamine (ED) and labelled as MOF-74(Zn)-ED. The fluorescence enhancement revealed a good linear relationship with the concentrations range of TBBPA from 50 to 400 μg/L, with LOD 0.75 μg/L. Moreover, Förster resonance energy transfer (FRET) could be ascribed to the possible mechanism of the fluorescence enhancement. The promising results obtained from this study could pave a way for the detection of other contaminants in the environment.

In recent years, more papers have used novel analytical techniques to determine TBBPA in environmental and food samples. This shows that these methods are replacing traditional detection methods. A summary of analytical techniques for TBBPA has been shown in Table 4.

### 7.3. Treatment Methods/Processes

A typical point source of TBBPA is WWTPs, however, they serve as significant treatment barriers for decreasing the rate at which pollutants are released from waste water into the environment. Over the years, treatments in a lab have been effective. Most researchers have employed chemical and physical methods, such as ozonation, adsorption, and oxidation; and biological methods, such as anaerobic degradation [204]. Chen and co-workers [205] focused on anaerobic co-metabolic biodegradation of TBBPA and a degradation rate of 93.2% was achieved with a short half-life of 12.81 h. However, the application of a non-biological treatment technique caused an increase in the degradation rate from 29.74 to 100% after 60 min ozonation [4] and in 40 min, 98% degradation was achieved by sorption [206]. Researchers have reported other treatment strategies such as UV-vis/Ag@TiO_2_ [207], UV oxidation [208], adsorption by nanoparticle composites [209,210]. Additionally, removal efficiencies in operating WWTP have yielded promising results in most countries. In Busan, Korea the removal efficiency of TBBPA was (64 ± 32%) [48]. Du’s group [211] observed that approximately 80% of TBBPA was removed from a printed circuit board (PCB) industrial waste under different laboratory conditions. By means of a novel bio-electrochemical technology, 95% TBBPA was removed in 120 min from activated sludge from a municipal WWTP in Shanghai, China, [185]. Since larger quantities of waste water are generated, more information on the leaching behavior of TBBPA is needed. The disposal of sludge should be carefully considered because it is a potential pathway for TBBPA to be released into the environment. The acidity of the landfill leachate limited leaching [115]. This is an indication that landfilling could be a suitable final disposal method.

Over the years, formal e-waste recycling facilities have made use of technologies for treating PCB and BFR plastics. These developments pertain to multiple steps such as: pretreatment, recovery, and purification. Processing of waste has become increasingly challenging due to the complexity of waste, so to recycle waste effectively, the correct combination of technologies is required [212]. In the recycling of PCB, it is economical to use high-temperatures for treatment. However, there are certain drawbacks, which include the emission of hazardous fumes (dioxins and furans) and generation of large volumes of slurry. 

Research has revealed that advancements made in the recycling of PCB are associated with the pretreatment process, which aims to enhance the release of the components and further increase the concentration of the precious metals. A recent innovation by Yang’s group [213], utilized an abrasive water jet to cut the hard structure of PCB instead of using shredders. The researchers employed a less energy-intensive technique to reduce the size of the PCB from 14 mm to less than 1 mm. Another recent development by Chu and co-workers [214] utilized heat to prepare the feed for enhanced recovery. The nickel and copper content of the PCBs were oxidized and the waste was heated at 600 °C. Hence, copper was highly selective after the bioleaching step (10% nickel and 100% copper was recovered) and the whole procedure was 22 times faster. To enhance the bioleaching method used to extract metals from PCB a group of researchers attempted to eliminate tin and lead from PCB using nitric acid. Afterward, they successfully bioleached more than 92% of nickel, copper, and zinc [215]. Another study explored a number of procedures to recover the organic component while reducing the expense of the usual pretreatment methods. The researchers recommended procedures, which include froth flotation, thermal shock, and pyrolysis, these methods do not involve rigorous crushing and size reduction. They also proposed an organic swelling procedure that entails extracting the waste organic components using organic solvents (dimethyl sulfoxide, N-methyl-2-pyrrolidone, and dimethylformamide) [216]. The study by Sousa et al. [217] is a clear example, they applied dimethyl sulfoxide and N-methyl-2-pyrrolidone at 180 °C for 30 min to completely separate metals from PCB.

The recovery methods that convert the waste to useful materials such as graphite, metals, and organics have advanced significantly in recent years. Recovery techniques (pyrometallurgical and hydrometallurgical processes) have been employed to recycle valuable elements from electronic waste [218,219]. With the application of hydrometallurgical procedures, metals are recovered via electrolysis, leaching, precipitation, and solvent extraction [220,221], resulting in highly purified products, low energy consumption, and low emission rates. Some of the recent innovations in PCB recycling is the use of new reagents (thiocyanate and thiosulfate) to leach precious metals [222,223] and the application of bioleaching has been a sustainable metal recovery technique [224,225]. In bioleaching, different microorganism (fungi, acidophilic bacteria, and cyanogenic bacteria) are utilized to generate the required reagents to leach metals from electronic waste [226]. The mechanism used by fungi is to convert sugars to organic acids, acidophilic bacteria either produces acid or convert Fe^2+^ to Fe^3+^ to serve as an oxidant. Moreover, amino acids are employed by the cyanogenic bacteria to produce cyanide [227,228]. The advantages of bioleaching generally include being economical and environmentally friendly, yet its shortcomings are low leaching efficiency (often less than 60%) and slower kinetics (on the order of weeks to months). Despite all of its advantages and advancements, hydrometallurgy remains an inefficient process for its application in diverse and complex materials, hence it is not feasible to be used as the first step [229,230]. Pyrometallurgy, on the other hand, is rapid, effective, and insensitive to the composition of the feed. It also does not involve complex pretreatment procedures and is appropriate for large-scale pretreatment of PCB [231]. The study by Heo et al. [232] is an innovation in the pyrometallurgical recycling of PCBs. In their work, they reduced PCB and then sprayed oxygen-rich air into a molten copper alloy. This resulted in the removal of most of the impurities.

Furthermore, thermal methods have been useful for hazardous waste (coal in power plants, municipal solid waste, sewage sludge) treatment. This process could be applied in the treatment of PCB. The recycling of PCB results in environmental contamination from BFR, organic plastics, and heavy metals. In general, organic bromide in flame retardants is transformed into HBr and Br_2_. Debromination prior to pretreatment is usually carried out to avoid the generation of brominated furans and dioxins in high-temperature recycling processes [233]. Most improvements in thermal methods for treating wastes are associated with boosting overall performance, recovering the non-metallic components, and minimizing emissions. For this purpose, processes such as pyrolysis, hydrogenation, gasification and incineration have been developed. In pyrolysis, the waste is heated without oxygen, which results in the transformation of the plastics to fuel gases, char, and oil; in hydrogenation, the plastics are exposed to temperature and pressure, then transformed to oil; in gasification, limited oxygen supply is applied in combustion and the gases produced are recovered as fuel (different hydrocarbons, H_2_, and CO); and in incineration, after combustion of materials, energy is recovered as heat [234]. Among all these thermal procedures, pyrolysis and controlled incineration have garnered considerable interest in the recycling of PCBs, because they can recover energy and the metallic components with lower emission levels [235,236]. Some of the innovations in this field include application of microwave-assisted pyrolysis, designing co-pyrolysis, developing better reactors, and introducing catalysts to the system to enhance the transformation of organic materials [237]. Correspondingly, the most practical strategy to recover plastic fractions from PCBs, involves energy recovery from plastics and chemically converting polymers in PCB [234].

## 8. Concluding Remarks and Perspectives

This review summarizes the human exposure pathways and bioaccumulation, human biomonitoring, toxicity studies and assessment, mechanism of action, environmental impact, emerging methods for detection and treatment of TBBPA in the environment. The effects of TBBPA on the environment, humans, and animals have been widely studied in different research groups and laboratories due to its toxicity. The ubiquitous nature of TBBPA has led to its occurrence in human samples and different environmental matrices. Previous studies on exposure routes in humans are preliminary and have several limitations. Therefore, further studies are warranted for the development of connections between TBBPA in a non-invasive matrix and its levels in blood to ensure that the non-invasive matrix accurately depicts the overall body load without interfering with levels of external exposure. 

In addition, thermal and integrated methods for TBBPA degradation are not widely applied, but they have the potential to be an appealing, and cost-effective strategy. Although membrane separation and photocatalytic oxidation methods have been combined in most studies, the combination of other techniques need to be considered and investigated. In addition, thoroughly exploring the binding of TBBPA in adsorbent-based treatment will be desirable as the mechanism is not clear. Even though it is feasible for TBBPA to degrade into a number of byproducts with increased toxicity and estrogen-binding activity, very little research has concentrated on removing the emerging organic contaminants from complex matrices. This could impede the development of TBBPA degrading technology. Hence, more research is required to determine the environmental trends relating to TBBPA.

## Figures and Tables

**Figure 1 molecules-28-02505-f001:**
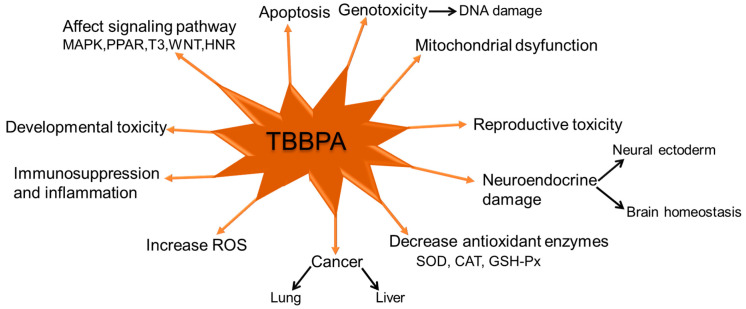
Graphical illustration of TBBPA induced toxicity. MAPK-mitogen activated protein kinase; PPAR-peroxisome proliferator-activated receptor; T3-triiodothyronine; WNT-wingless/integrated; HNR-Heinz Nixdorf recall; SOD-superoxide dismutase; CAT-catalase; GSH-Px-glutathione peroxidase; ROS-reactive oxygen species.

**Figure 2 molecules-28-02505-f002:**
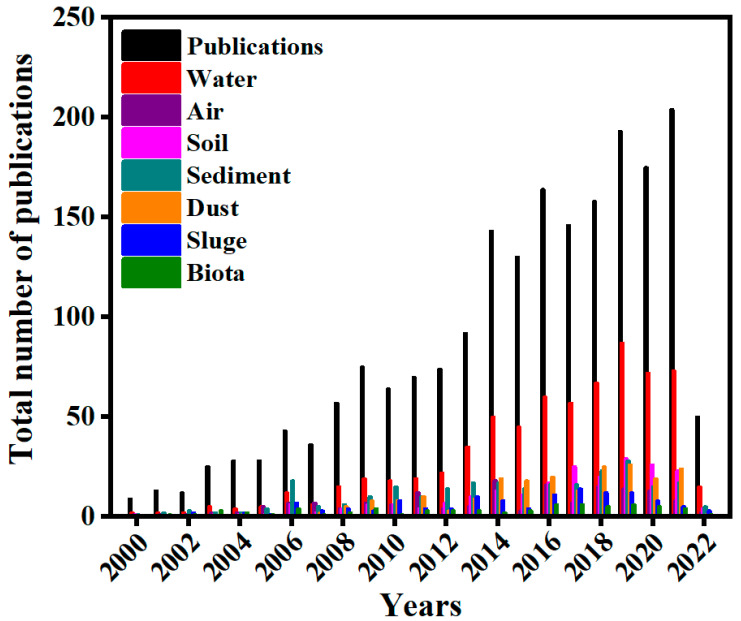
Yearly publications on TBBPA monitoring in environmental samples (Web of Science as of 4–4-2022).

**Figure 3 molecules-28-02505-f003:**
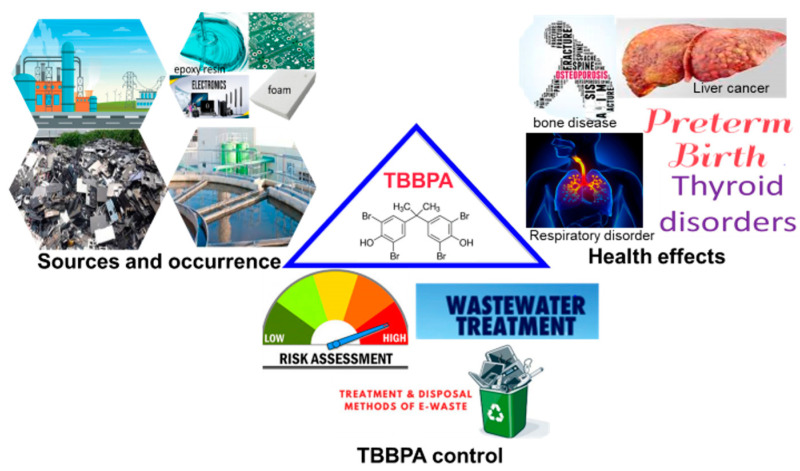
Schematic illustration of the sources and occurrence of TBBPA in the environment, health effects and control.

**Table 1 molecules-28-02505-t001:** Detected concentrations of TBBPA in human samples.

Scheme	Number of Samples	Concentration	Reference
Breast milk	1600	0.14–66.8 ng/g lw	[19]
	64	<0.1–2200 ng/g lw	[20]
	111	<LOD-42 ng/g lw	[57]
	106	<LOD-15.1 ng/g lw	[61]
	53	-	[63]
	120	0.03–0.17 ng/g lw	[64]
	11	<0.29–0.17 ng/g lw	[65]
	-	4.73 ng/g lw	[66]
	50	<2–688 ng/g lw	[67]
	36	<0.04–0.65 ng/g lw	[68]
	43	<0.03–0.55 ng/g lw	[69]
	19	ND – 8.7 ng/g lw	[70]
Hair	15	ND-1.08 ng/g	[53]
	24	ND-16.04 ng/g	[53]
	-	ND-16.04 ng/g	[53]
Serum	76	8.61–83 ng/g	[48]
	-	≥480 ng/L	[49]
Urine	140	0.19–127.24 µg/g	[42]
	86	0.0793 µg/L	[43]
	104	0.633 µg/L	[43]
	110	1.15 µg/L	[43]
	40	0.030–0.830 µg/L	[71]

ND-not detected; LOD-limit of detection.

**Table 2 molecules-28-02505-t002:** Concentrations of TBBPA in water, dust, air, sediment and soil.

Matrices	Sampling Year	Concentration	Reference
Sea water	2015	ND-460 ng/L	[21]
River water	2017	ND-12.279 ng/L	[119]
Lake water	2018	18.5–82.6 ng/L	[120]
River	2018	18.5–82.6 ng/L	[120]
River water	2013	ND-920 ng/L	[135]
Surface water	2014	ND-32.3 ng/L	[136]
Road dust	2016	<LOD-74.1 ng/g dw	[125]
Indoor dust	2017	ND-144 pg/m3	[137]
Outdoor dust	2017	ND-326 pg/m3	[137]
Office dust	2016–2017	<0.1 pg/m3	[138]
House dust	2011	69 ng/g dw	[139]
Sediment	2019	19.8–1.52 × 104 ng/g dw	[34]
	2017	ND-3.889 ng/g dw	[119]
	2019	0.76–2.51 ng/g dw	[121]
	2020	108–3.1 × 103 ng/g dw	[121]
	2012	0.003–0.31 ng/g dw	[122]
	2019–2020	0.02–21.5 ng/g dw	[123]
	2013	ND-230 ng/g dw	[135]
	2015	0.168–2.66 ng/g dw	[136]
	2009–2010	0.06–300 ng/g dw	[140]
Biota	2019	6.96–1.97 × 105 ng/g ww	[34]
	2013	0.56–22.1 ng/g ww	[122]
	2019	<LOD-42.8 ng/g ww	[124]
Soil	2016	<LOD-33.8 ng/g dw	[125]
		4.4 ng/g dw	[126]
		0.025–78.6 ng/g dw	[126]

ND-not detected; LOD-limit of detection.

**Table 3 molecules-28-02505-t003:** Sample pretreatment methods for TBBPA detection.

Sample Type	Pretreatment Method	Main Components	Detection Device	Limit of Detection	Reference
				ng/g	
Sea food	QuEChERS-LLE	hexane:tertbutylmethylether	LC-MS/MS	0.006	[141]
		(3:1 *v*/*v*), NaCl, MgSO_4_			
Sediment	MAE-QuEChERS	Tetrahydrofuran: methanol (50:50, *v*/*v*)	UHPLC-MS/MS	0.1 and 0.5	[142]
Sludge	MIP-SPE	Methanol, water	HPLC-MS/MS	0.0004–8.28	[144]
Soil	MAE-LLE	-	HPLC-MS	1.4	[145]
				µg/L	
Water	MC-LLME	-	HPLC	0.16–0.23	[143]
Plastic waste	MAE	Isopropanol/hexane; hexane/ethanol	GC/MS	-	[147]
Serum	UAE-QuEChERS	Ethyl acetate:hexane (75:25, *v*/*v*)	UHPLC-MS/MS	0.1 and 1.0	[148]
Dust	DL-LME	acetone/methanol (75:25, *v*/*v*), acetonitrile, ethanol, and 2-propanol	GC/MC	-	[149]
Water	M-SPE	Methanol, magnetic nanoparticle	HPLC	0.011	[150]
Water	SPE	Methanol:acetonitrile (1:1, v:v)	UPLC-MS/MS	0.0001	[151]
Water	DSPE	MoS_2_/CD:acetonitrile	HPLC	0.01–0.06	[152]
Water	MSPE	Methanol, acetonitrile	GC-MS	0.13	[153]
Water	SPME	-	CFDI-MS	0.0001–0.0032	[155]
Water	DMIP	DIP, BPA, methanol	RRLC	2	[156]
Sewage	MIP-SPE	Methanol, water	HPLC-MS/MS	7 × 10^−7^–0.0163	[157]

RRLC: rapid resolution liquid chromatography; CFDI-MS: constant flow desorption ionization-mass spectrometry.

**Table 4 molecules-28-02505-t004:** Analytical techniques for TBBPA detection.

Matrix Type	Detection Method	Detection Limit	µg/L	µg/g	Reference
Water	UPLC-MS	0.1–2.5 ng/L	0.0001–0.0025		[151]
	EC/I	0.1 ng/mL	0.1		[178]
Waste water	EC/S	25 pM			[179]
Soil	UPLC-MS/MS	0.22–88 pg/gdw		2 × 10^−7^–8.8 × 10^−5^	[180]
	FELISA	0.05ng/g		5 × 10^−5^	[181]
Sewage	ELISA	<LOD-0.75 ng/mL	<LOD-0.75		[182]
Sludge	ELISA	<LOD-3.65 ng/mL	<LOD-3.65		[182]
landfill leachate	ELISA	<LOD-1.17 ng/mL	<LOD-1.17		[182]
Dust	MIP-CO	3 pg/g		3 × 10^−6^	[183]
Indoor dust	MIPES	0.5 nM			[197]
Beverage	EC/S	0.75 nM	0.00075		[198]
Food samples	RFI	0.118 µg/L	0.118		[199]
Plastic products	EC/S-MIP	0.015 nM			[200]
E-waste	FL-MIP	3.6 ng/g		0.0036	[201]
Plastic e-waste	FL-MIP	5.4 nM			[202]
Sediment	BLEIA	2.5 pg/mL	0.0025		[203]
UPLC-MS-ultra-high performance liquid chromatography-mass spectrometry					
EC/I-Electrochemical immunoassay; EC/S-Electrochemical sensor					
FELISA-Fluorescence enzyme linked immunosorbent assay					
ELISA- Enzyme linked immunosorbent assay					
BLEIA-Bioluminescent enzyme immunoassay					
RFI-Ratiometric fluorescence immunoassay					
MIP-Molecular Imprinted Polymer					
MIP-CO-Molecular imprinted polymer-colorimetry					
MIPES-Molecular imprinted polymer photoelectrochemical sensor					
FL-MIP-Fluorescence-molecular imprinted polymer					

## Data Availability

Not applicable.

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
