# Peer review of "A Review on Tetrabromobisphenol A: Human Biomonitoring, Toxicity, Detection and Treatment in the Environment"

_molecules, 2023, doi:10.3390/molecules28062505_

Round 1

Reviewer 1 Report

Recommendation: Major Revision

The review submitted to molecules “A review on Tetrabromobisphenol A: human biomonitoring, toxicity, detection and treatment in the environment”. Overall, the review paper looks useful, but still, several concerns need to be resolved. Therefore, I recommend this manuscript for publication after Major Revision. The following comments need to be considered before resubmission.

Comments:

1.      The is general, the author should revise the abstract

2.     The author must double check the reference format in the text.

3.     The are several grammatical errors, the author should proofread the manuscript with a native English speaker to avoid grammatical and typo errors.

4.     The author must improve the conclusion part by addressing the main points.

5.     The author must include the latest relevant references 

Author Response

Point 1: The is general, the author should revise the abstract

Response 1: Thank you for pointing this issue out, the abstract has been carefully revised and changes highlighted red.

Point 2: The author must double check the reference format in the text.

Response 2: We agree with this suggestion and have modified the reference format according to journal requirement.

Point 3: The are several grammatical errors, the author should proofread the manuscript with a native English speaker to avoid grammatical and typo errors

Response 3: We apologize for grammatical and typo errors. We worked on the manuscript for a long time and the repeated addition and removal of sentences and sections obviously led to the errors. We have now worked on the grammatical errors and typos. We really hope that the errors have improved.

Point 4: The author must improve the conclusion part by addressing the main points.

Response 4: Thank you for your good suggestion. The conclusion section has been improved and the main points addressed and highlighted red in the revised manuscript.

Point 5: The author must include the latest relevant references 

Response 5: Thank you for your comment. Thomsen et al., 2001 has been replaced with Li et al., 2017; Reistad et al 2005 has been replaced with Yu et al 2019.

Reviewer 2 Report

The manuscript is well organized and does an extensive review of the evidence on biomonitoring and effects of TBBPA, and on pretreatment and detection methods. 

Because the concentrations in different papers do not use the very same units the authors in the text do not use a congruent notation, e.g. ng/g or ng g-1. I suggest adopting just one format.

Some of the values of concentration are given in wet weight and others in lipid weight. Also in this case the indication is not uniform and changes along the text.  I think that an introduction of a paragraph in the final part of the introduction resuming the notation could also help the reading and understanding of the values compiled.

I think that also in some sentences the indication of the references could be more direct. For example in lines 1047 to 1049, the way you present Sousa et al., 2022 seems a double indexation. The sentence could be just:  Th study of Sousa et al. (2022) is clear ... avoiding to use in the (Sousa et al 2022).

 The paper is well written, but there are some issues that deserve attention:

1.      In the final part should be included a paragraph explaining the units  used:

The concentrations are presented in ng/g wet weight (ww),  ng/g dry weight (dw), ng/g  lipid weight (lw), and ng/L depending on as the studies reported were run;

2.      The acronyms  adopted should be  controlled and verified because some are explained well after the first time are used and some are not defined;

3.      In the tables the authors tend to present the values as in the original paper, making it difficult to read and compare. Because the experiments the field measurements are not made in some conditions, eventually is not possible to give all the values in the same units, but is important to use congruent and uniform criteria. If authors think it´s important to preserve the original form, I suggest adding a new column  with values convert to be compared;

4.      The way the references are signed in the paper should be fully controlled; 

5.      The final part of the paper does not include the items: Author contributions; Funding and Conflicts of interest;

6.       The references have a general problem just the first author is presented. So the other authors must be added according to we the edition policy of the journal.  References 49 and 50 must be completed and as far I noted are missing 167a: Sastre (2005)

A.M. Sastre, J. Szymanowski,

EXTRACTION | Solvent Extraction Principles,

Editor(s): Paul Worsfold, Alan Townshend, Colin Poole,

Encyclopedia of Analytical Science (Second Edition),

Elsevier,

2005,

Pages 569-577,

ISBN 9780123693976,

https://doi.org/10.1016/B0-12-369397-7/00688-9.

(https://www.sciencedirect.com/science/article/pii/B0123693977006889)

 and 189a:     Tay et al., 2019

J.H. Tay, U. Sellström, E. Papadopoulou, J.A. Padilla-Sánchez, L.S. Haug, C.A. de Wit

Serum concentrations of legacy and emerging halogenated flame retardants in a Norwegian cohort: relationship to external exposure

Environ. Res., 178 (2019), Article 108731, 10.1016/j.envres.2019.108731

 Thank you in advance for your attention. I look forward to hearing from you.

A complete list of notes and comments can be found in the pdf file attached

Author Response

Point 1: In the final part should be included a paragraph explaining the units used:

The concentrations are presented in ng/g wet weight (ww), ng/g dry weight (dw), ng/g lipid weight (lw), and ng/L depending on as the studies reported were run;

Response 1: Thank you for your useful comment. The full meaning of the units used has been highlighted red in line 1132-1136.

Point 2: The acronyms  adopted should be  controlled and verified because some are explained well after the first time are used and some are not defined;

Response 2: We agree with this suggestion and have explained the acronyms that have not been defined. These acronyms have been highlighted red on line 68,304,392,507,540,783,784,785,793,824,826,857,985.

Point 3: In the tables the authors tend to present the values as in the original paper, making it difficult to read and compare. Because the experiments the field measurements are not made in some conditions, eventually is not possible to give all the values in the same units, but is important to use congruent and uniform criteria. If authors think it´s important to preserve the original form, I suggest adding a new column  with values convert to be compared;

Response 3: Thank you for your suggestion. We have used consistent and uniform units making it easy to read and compare.

From Table 1: all breast milk samples are represented in ng/g/lw; hair ng/g; serum ng/g; urine µg/L.

From Table 2: the unit for sea water was changed from ND-0.46 µg/L to ND-460 ng/L, so all the water concentrations are in ng/L; concentration of sediments and soil are  presented in ng/g dw; biota concentrations are presented in ng/g ww.

From Table 3: the concentrations of soil, sludge, sediment and sea food are represented in ng/g and water, sewage and serum concentrations are represented in µg/L

From Table 4: different columns were created and labelled as µg/L and µg/g ,units were converted to suit the category labelled.                     

Point 4: The way the references are signed in the paper should be fully controlled; 

Response 4: Thank you for your insightful comment. The references cited in the manuscript has been controlled and changed according to the required format.

Point 5:The final part of the paper does not include the items: Author contributions; Funding and Conflicts of interest;

Response 5: Thank you for the good comment. The author contributions; funding and conflicts of interest have been included at the end of the manuscript on page 49 and highlighted red.

Point 6: The references have a general problem just the first author is presented. So the other authors must be added according to we the edition policy of the journal.  References 49 and 50 must be completed and as far I noted are missing 167a: Sastre (2005)

Response 6: Thank you for your helpful comment, we apologize for the error. According to your suggestion the references have been represented according to the journal format. Reference 49 and 50 are now represented as [14] and [45 ], respectively, because of the change in reference format. The link to EFSA has been included. An updated reference of A.M sastre has been included, which is now [150]. Tay et al.2019 has also been added it is represented as [47].

Round 2

Reviewer 1 Report

Accept

Reviewer 2 Report

I think that the manuscript was very much improved with the modification proposed. I noted some typing problems that should be corrected, plese see the file.

The quality of figure 2 also could be improved.
